# Physical Limitations on Fundamental Efficiency of SET-Based Brownian Circuits

**DOI:** 10.3390/e23040406

**Published:** 2021-03-30

**Authors:** İlke Ercan, Zeynep Duygu Sütgöl, Faik Ozan Özhan

**Affiliations:** 1Department of Microelectronics, Delft University of Technology, 2628 CD Delft, The Netherlands; 2Electrical and Electronics Engineering Department, Boğaziçi University, İstanbul 34342, Turkey; zeynep.sutgol@boun.edu.tr (Z.D.S.); faikozan.ozhan@boun.edu.tr (F.O.Ö.)

**Keywords:** Brownian circuits, SET transistors, fundamental bounds

## Abstract

Brownian circuits are based on a novel computing approach that exploits quantum fluctuations to increase the efficiency of information processing in nanoelectronic paradigms. This emerging architecture is based on Brownian cellular automata, where signals propagate randomly, driven by local transition rules, and can be made to be computationally universal. The design aims to efficiently and reliably perform primitive logic operations in the presence of noise and fluctuations; therefore, a Single Electron Transistor (SET) device is proposed to be the most appropriate technology-base to realize these circuits, as it supports the representation of signals that are token-based and subject to fluctuations due to the underlying tunneling mechanism of electric charge. In this paper, we study the physical limitations on the energy efficiency of the Single-Electron Transistor (SET)-based Brownian circuit elements proposed by Peper et al. using SIMON 2.0 simulations. We also present a novel two-bit sort circuit designed using Brownian circuit primitives, and illustrate how circuit parameters and temperature affect the fundamental energy-efficiency limitations of SET-based realizations. The fundamental lower bounds are obtained using a physical-information-theoretic approach under idealized conditions and are compared against SIMON 2.0 simulations. Our results illustrate the advantages of Brownian circuits and the physical limitations imposed on their SET-realizations.

## 1. Introduction

The design of future computers focuses on minimizing fluctuations, as they are perceived as an impediment to be avoided at any cost. A wide range of computing proposals are presented which utilize noise for increased performance efficiency in computing [1,2,3,4,5,6,7]. Among these novel approaches, an emerging paradigm stands out as it implements Brownian motion in electronic circuit architectures that can exploit fluctuations to reduce energy dissipation in computing [8,9,10]. Brownian circuit technology is designed to efficiently and reliably perform primitive logic operations in the presence of noise and fluctuations, and Single-Electron Transistor (SET) technology is proposed as an appropriate technology-base to realize these circuits, as it has been studied extensively and meets the conditions needed to process token-based signals that are subject to fluctuations [9]. Other technology bases, such as nanophotonics or skyrmions, are proposed as suitable alternatives [11]. The foundations of Brownian circuits, along with their potential realization via SET technology, is studied in a wider context in the existing literature. However, the efficiency limitations of this technology proposal and its performance against existing paradigms has still not been completely revealed.

There is an ongoing effort to assess the fundamental limitations of novel circuit paradigms, to help assess upcoming trends in the future of computing [12,13,14,15,16,17,18,19,20,21,22,23,24,25,26]. Recent experiments show that we can obtain energy dissipation in the $k_{B}T$-level [27]. However, in order to calculate the fundamental efficiency limitations of emerging computing paradigms, we need foundational approaches that allow for the incorporation of intricate details of the circuit operation and physical structure. There is an attempt to treat nanoelectronic circuits using a physical-information-theoretic framework, where the circuit and its surroundings are treated as composite quantum systems. These analyses provide us with fundamental lower bounds on the performance efficiency of post-CMOS technologies.

The fundamental lower bounds of a given technology-base allow us to obtain best-case scenarios of performance projections based on the underlying computing strategy by taking the inefficiencies that are imposed by the nature of computing paradigms into account. In our earlier work, we laid out the foundations of a methodology to obtain fundamental efficiency limits for complex computing structures and calculated the fundamental lower bounds on energy dissipation at the circuit level for non-transistor-based [15,16], transistor-based nanoelectronic technology proposals [17], and optical microring resonator-based logic circuits [24,25], as well as at the nanoprocessor level [18,26]. The approach has also been implemented in finite-state machines [19]. Such a methodology, providing paradigm-dependent fundamental energy dissipation bounds, is a powerful tool, which can be used to obtain performance projections of post-CMOS devices and provide valuable insight into the upcoming challenges in emerging technologies [20,28,29]. Earlier applications of our methodology consist of the study of fundamental limitations of emerging nanoelectronic technologies with low energy dissipation, including a Brownian half adder designed by Peper et al. in [23] under idealized conditions.

In this paper, we study physical conditions that affect the efficiency of Brownian circuits and discuss the parameters for SET-based implementations from a fundamental point of view. We first provide an overview of SET-based Brownian circuit primitives and study the physical conditions that affect their efficiency using SIMON 2.0 simulation results. Then, we present a Brownian two-bit sort circuit design and illustrate the fundamental lower bounds on energy dissipation associated with information processing in these circuits. We compare our findings with the simulation results obtained in SIMON 2.0 and show how the additional effects incorporated in the bound capture the essential functional features of the circuit further, before concluding with our final remarks.

## 2. Simulating SET-Based Brownian Circuit Primitives on SIMON 2.0

### 2.1. Fundamentals

Brownian circuits are proposed by Peper et al. as a novel approach that utilize noise and fluctuations in the circuit in order to find computational paths through a random search, and therefore improve the efficiency of information processing. These circuits represents signals in terms of tokens—positive charges—propagating randomly in the network. The building blocks of Brownian circuits are Ratchet, Hub and Conservative Join (CJoin). Token flow in a Ratchet is unidirectional, whereas the transitions in Hub and CJoin are bidirectional. Figure 1 illustrates the direction of token flow in these circuit primitives based on the description provided in the literature. As compared to Hub and CJoin primitives, Ratchet performs asymmetric operation, allowing tokens to flow only in the forward direction; tokens are not allowed to tunnel backwards, which leads to deadlocks, a phenomenon commonly seen in conventional token-based circuits. Based on the functional description of these primitives, for a technology to be suitable for Brownian circuit implementations, it is necessary for it to support the representation of signals that are token-based and subject to fluctuations. If these conditions are met, then the universal logic operations can be obtained based on the transition rules associated with each element, and hence can be realized as illustrated in the SET transistor application. Further details on local transition rules used to drive Brownian circuit primitives are provided in [9].

An SET transistor is composed of a conducting island or quantum dot, and two electrodes known as the drain and the source, connected through tunnel junctions to one common electrode with a low capacitance, the island. The electrical potential of the island can be controlled by a third electrode, the gate, which is capacitively coupled to the island. Since electron tunneling is a quantum mechanical process, the wave function of an electron expands over potential barriers of tunneling junctions, and the electron is circulated in an SET circuit. If this effect was dominant, there would be no controlled charges, and computations by using localized electrons would not be possible. In order to obtain a discretized charge of an electron on each island, the tunneling junctions should have a high tunneling resistance. Moreover, the charging energy (Coulomb energy) must dominate over the quantum fluctuations. This condition can be expressed as [9]
$$\left(q_{e}^{2}/2C_{j}\right).R_{j}.C_{j}>>h=>R_{j}>>h/q_{e}^{2}=25.8\;k\Omega$$
*h* is the Planck’s constant, $C_{j}$ is the tunneling capacitance, and $R_{j}$ is the tunneling resistance. Thus, the resistance of all the tunneling junctions in recent studies is taken as 100 kΩ [9]. Another condition which has to be considered is the thermal energy. If the thermal energy is greater than the Coulomb energy in an SET circuit, $E_{c}$, the quantum tunneling effects become ineffective. The condition can be expressed as
$$E_{c}=q_{e}^{2}/2C>>k_{B}T$$
$k_{B}$ is Boltzmann’s constant and *T* is the temperature. In order to realize Brownian circuits in SET transistors, we need to use the token-based character of electrons by making the tunneling resistance and capacitance values convenient with these conditions.

The circuit parameters for the SET implementation of Brownian circuit primitives are optimized for operation at T=1 K in [30]. In an effort to understand the impact of physical circuit parameters on the operation of these primitives, we studied each circuit element under varying temperatures. We used SIMON 2.0 to illustrate the circuit behaviour for different parameters. There are three fundamental approaches to the simulation of single-electron circuits: the Monte Carlo method, SPICE macromodeling and solving Master Equation [31,32]. SIMON 2.0 is a single-electron tunnelling device and circuit simulator that is based on a Monte Carlo method. The program provides a transient and stationary simulation of SET-based circuits consisting of tunnelling junctions, capacitors, and voltage sources. SIMON 2.0 has a graphical user interface and a circuit editor. It is mostly used to understand the behavior of the Coulomb blockade and SET oscillations. Below, we present the SIMON 2.0 simulation results obtained for Brownian circuit primitives.

### 2.2. Brownian Ratchet

Brownian Ratchet is a deterministic circuit primitive and the simplest building block of Brownian circuits. It is also linked to the Maxwell’s demon since it enables deterministic computation by indeterministic means, as illustrated by Serreli et al. [33]. SET implementation of a racthet circuit primitive is similar to that of a single SET structure with a different capacitative coupling value. The source and drain, included in Ratchet, are capacitatively coupled to electrodes. Thus, source and drain potentials can be controlled, just like the island. The deterministic nature of the Ratchet distinguishes it from other circuit primitives as its operation does not involve Brownian search. Brownian search is a random process and included in other two building blocks, Hub and CJoin, which significantly affects the stability of these circuit primitives in high temperatures. The simulations involving Hub and CJoin circuit elements involve Brownian search and are therefore probabilistic in nature.

In an effort to study the impact of temperature on Ratchet’s behaviour, we illustrated the tunneling pattern for different junction capacitances at varying temperatures, as shown in Figure 2. At $V_{2}=30$ mV, we change the junction capacitance from $J_{1}=J_{2}=10^{-17}$ F (black line), to $J_{1}=J_{2}=10^{-18}$ F (red line). For $J_{1}=J_{2}=\;10^{-18}$ F, the maximum temperature of the Ratchet can function can reach 8.125 K as desired. Increasing the junction capacitances, however, also has an impact on the circuit stability.

The stability plots for SET transistors are obtained in SIMON 2.0 by determining the regions where Coulomb blockade occurs, as shown in [34,35]. These figures are often plotted with respect to the gate voltage and bias voltage as shown in Figure 3. A Brownian Ratchet is simple a SET circuit with additional capacitors on either side, $C_{1}$ and $C_{3}$ capacitors. We replicated stability plots for a Brownian racthet in Figure 3 with the following circuit parameters $C_{1}=C_{2}=C_{3}=3\times 10^{-18}$ F, $J_{1}=J_{2}=3\times 10^{-17}$ F and $R_{1}=R_{2}=10^{5}\;\Omega$, where $V_{1}$ and $V_{2}$ correspond to the bias and gate voltage, respectively. The white diamond shaped regions mark the areas where the Coulomb blockade holds, referring to the bias values where the number of charges in the island is fixed and no current flows. The dark lines designate the values where the number of electrons fluctuate and the current flows, due to its overcoming the coulomb blockade. The size of the diamond depends of the ratio between the capacitance values of the junction and the capacitors.

We observed that as temperature increases, the unstable region dominates over the stable region. However, our simulations on the affect of junction capacitance on Coulomb blockade regions illustrate that we can choose appropriate circuit parameters at a given temperature that will accommodate the desired tunneling behaviour for the Brownian circuits often obtained at the boundary between these regions.

We also observed the affect of temperature on tunneling behaviour. Figure 4 illustrates the impact of temperature increase on current. Our findings are in line with Figure 9 of [31] presented by Wasshuber et al. The discrete tunneling behaviour seems to be maintained at relatively higher temperatures; however, given the stability patterns, the desired Ratchet circuit behaviour is observed to deteriorate above 5K. Therefore, we restricted our observations to lower temperatures.

We also study the impact of bias voltage on the tunneling events. SIMON 2.0 is a Monte Carlo based simulator, where the obtained results depend heavily on the number of events, which include a range of probabilistic scenarios. We set the event number sufficiently highly, to obtain a consistent picture of the circuit behavior. We can see that, for a given Ratchet circuit element implemented using SET transistors, for the same bias voltage, the average timepoint at which tunneling occurs remains fixed; however, the time interval marked by the first and last tunneling events included in the probabilistic scenarios of the Monte Carlo simulation increases with temperature. In addition to the temperature, increasing the bias voltage causes the first tunneling event to occur at an earlier point in time, i.e., the temperature increases the probability of the tunneling events taking place at an earlier point in time. These findings confirm that by increasing the bias voltage, tunneling events may occur at an earlier point in time. This also suggests that, for higher temperatures, $V_{bias}$ needs to be increased to avoid prolonged computation times. In Figure 5, we show the time at which the first tunneling event occurs in a Ratchet element at different temperatures and bias voltages. We show that the first tunneling event takes place at an earlier time point for higher voltages, and this initial tunneling event time occurs even earlier at higher temperatures.

In addition to shifting the time of the first tunneling event, increasing the bias voltage also increases the frequency of tunneling. Furthermore, the bias voltage required for tunneling to occur is fixed; for instance, for T=1 K, $V_{bias}$ is 15 mV and for T=2 K $V_{bias}$ is 21 mV. The minimum voltage for a higher tunneling probability increases with temperature as expected.

### 2.3. Brownian Hub

The hub circuit primitive is more complex in nature than the Ratchet element, as it incorporates Brownian search into its operation. When a token arrives at the input port of the Hub, it fluctuates between the three bidirectional ports, each of which can equally serve as output nodes, until the token is recognized as an input token by the following module, connected to the hub in one of the aforementioned ports [9]. The circuit parameters for the SET implementation of Hub circuit element are optimized for operation at T=1 K of Hub, as given in [30]. Operation of hte Hub physically starts when the input voltage exceeds a certain value and forces an electron to tunnel away from the first input node, creating a token. The omission of an electron at the node in question then forces another electron to tunnel from one of the other nodes to the input node and the token starts fluctuating in this manner. Figure 6 depicts the schematic of an SET-based Hub circuit. Hub circuit element can be described as follows: If $V_{a}$ is high, an electron departs from $n_{1}$ and tunnels through $J_{2}$ and $J_{1}$. Since there is a vacancy on the node $n_{1}$, the junctions $J_{3}$, $J_{4}$ or $J_{5}$, $J_{6}$ supply an electron to node $n_{1}$, from $n_{2}$ or $n_{3}$, respectively. On occasion, once $C_{s3}$ reaches a low enough value, the thermal energy of the electron is enough to overcome the voltage differences and the electron tunnels back to the node $n_{2}$ or $n_{3}$. Thus, the electron randomly jumps from the node $n_{1}$ to $n_{2}$ or $n_{3}$ and then back to $n_{1}$. In order to obtained the desired functionality, the capacitance and resistance values for the tunnel junctions in a Brownian hub are given in the literature as $C_{1}=C_{3}=C_{5}=C_{g}=C_{s1}=10\;\mathrm{aF},$
$C_{2}=C_{4}=C_{6}=0.1\;\mathrm{aF},$
$C_{s2}=0.5\;\mathrm{aF},$
$C_{s3}=0.2\;\mathrm{aF},$ and $R_{1}=R_{2}=R_{3}=R_{4}=R_{5}=R_{6}=100$ kΩ, respectively, and $V_{a}=V_{s}=16$ mV [9].

Unlike Ratchet, the default operation of the Hub requires it to remain unstable; therefore, stability plots in SIMON 2.0 do not provide any insightful information regarding its operation. The parameters that mainly control how input token is provided are the input voltage $V_{a}$ and input capacitance $C_{s2}$ values. Our simulations show that, with the parameters specified in [30], a Brownian search can only be conducted at a temperature of T=1 K. The operating temperature increases the values of $V_{a}$, and/or $C_{s2}$ must be increased for the same behaviour to be replicated at T=1 K. However, when we increase the value of the temperature from T=8 K to T=9 K, simulations showed that the Hub is deemed incapable of executing Brownian search regardless of the values of $V_{a}$ and $C_{s2}$. Changing the different capacitance values of the circuit may allow the Hub to operate at higher temperatures values; however, it remains as an unexplored territory in our research. Our research concludes that any Brownian circuit containing an SET implementation of the Hub primitive cannot operate above T=8 K.

### 2.4. Brownian CJoin

The CJoin circuit primitive moves the tokens on its two input lines to the two output lines in a pairwise manner. Unlike the Ratchet element, the tokens at the output of a CJoin can move in the reverse direction, making CJoin bidirectional in operation. For two input tokens to be processed by the CJoin element, they have to arrive simultaneously. Tokens arriving at the inputs asynchronously are not operated by the CJoin circuit primitive and may, therefore, fluctuate along the input–output lines. In the literature, CJoins are designed to operate in unidirectional or bidirectional manner. The original bidirectional CJoin design in [36] is improved so it can operate unidirectionally, in order to tackle routing problems and requirements for unrealistically small capacitance values in [30]. CJoins are used in conjunction with Hubs to obtain Boolean expressions [9]. The existing SET transistor implementations of the redesigned CJoin are optimized for 1 K temperature. Our simulations in SIMON 2.0 based on this design have proven to exhibit stable behaviour up to 6K temperature. Between 0.7 K and 1.5 K temperature, CJoin operates without charge fluctuations at the output. As the temperature increases, these fluctuations become dominant, and the net charge values at the output nodes does not reach $1q_{e}$, as shown in Figure 7. An increase in temperature effects CJoin stability, such that, beyond 2.1 K temperature, the charge is transported to the output lines, even when there is no supply in the input lines. CJoin operation starts with electron tunneling at the junctions that are connected to the quantum island in the input. Increasing the input capacitance on this node pulls the temperature range of CJoin functionality upwards. SIMON 2.0 simulations also showed that the time delay between two input tokens is decisive for the pairwise behavior of CJoin. Above a certain time duration, CJoin no longer waits for the second token to arrive before an already present token is moved to the output line of that branch. This duration is correlated to the temperature, such that, at 1 K, the input delay can be as long as 50 miliseconds; while the temperature increases up to 1.5 K, the inputs need to arrive at the nodes almost simultaneously for the circuit to maintain the CJoin functionality.

## 3. An Illustrative Example: Brownian Two-Bit Sort

We use the aforementioned Brownian circuit primitives to design a Brownian two-bit sort. Figure 8 depicts a Brownian two-bit sort constructed from four CJoins, eight Hubs and twelve Ratchets. The four CJoins (squares) and nearest neighboring Hubs (circles), along with associated Ratchets (triangles), form a 2×2 CJoin structure, shown by a small square box with a cross inside in [9].

The inputs of the two-bit sort are represented by 0s and 1s at the top and the right, according to dual-rail encoding. As shown in Figure 8 and the associated truth table in Table 1, each input is presented to the circuit with its complement. The work of the 2×2 CJoin, i.e., the four CJoins (squares) and nearest neighboring Hubs (circles), along with associated Ratchets (triangles), is to generate four minterms which will be used by Hubs to obtain the desired functionality. Hubs combine the minterms and produce outputs. However, output $O_{1}$ is produced from the top-right CJoin and output $O_{2}$ is produced from the bottom-left CJoin.

The circuit elements that are involved in computation vary depending on the input, i.e., only junctions along the path leading to the associated output are activated, leaving the junctions that are not on the path to the output inactive. Some of the constituent parts of the circuit work for each input configuration. When we apply input A=B=1, four Ratchets, only the bottom left CJoin and two Hubs are activated to obtain the output $O_{1}=O_{2}=1$. A Ratchet has two tunneling junctions, a Hub has six tunneling junctions and a CJoin has fourteen tunneling junctions [9]. Although the whole circuit has 112 tunneling junctions, some of them are used in a logic operation.

## 4. Energy Dissipation Analyses

In our approach, identifying the particle supply cost associated with the processing of an input has proven to be crucial to accurately determine the lower bound of the energy dissipated in computation. The bounds obtained using our approach are radically different than that of Landauer’s [37,38], as our methodology is rooted in quantum dynamics and thermodynamics [16]. We also introduce a referent system that holds the memory of initial input that is presented to the information processing artifact during computation. This fictitious system allows us to identify any logically irreversible loss of information as a result of computation by calculating the loss of correlation between these two systems, i.e., the referent system provides us with a reference of how much information is irreversibly lost during computation.

Figure 9 depicts the globally closed and isolated universe in which the physical information processing artifact, i.e., the circuit of interest, and its surrounding subsystems are situated. This physical abstraction allows us to incorporate all of the interactions that take place in a complete computational cycle of a single input. This physical abstraction enables us to assume unitarily evolution via Schrödinger’s equation. However, it is important to note that we do not solve the Schrödinger’s equation or define a Hamiltonian for the system; the state transformations of the density matrix based on the composite quantum system description shown in Figure 9 allow us to obtain energy and von Neumann entropy changes in the complete system.

In addition to the physical abstraction of the computing structure, we also develop a process abstractions. The referent system, R, is a physical system that keeps track of the input string and allows us to identify the amount of irreversible information loss as a result of computation without ambiguity. The separation of the complete universe into two domains, computational and environmental, also distinguishes this irreversible loss through the thermalization and resetting of the information-processing artifact via interaction with the environment. Here, the bath, B, is a heath bath in direct thermal contact with the artifact and is nominally at temperature *T*, and the environment includes reservoirs that “rethermalize” B.

The departure point of our calculations is the fundamental lower bounds on energy dissipated into the bath as a result of computation, which are obtained using the generalized *L*-machine approach for quantum systems, introduced by Anderson in [39]. The bound given in Equation (66) of [39] is applied to nanoelectronic circuits in [16]. Below, we expand the bound obtained in [16] and incorporate the effect of non-idealized circuit operation in SET, such as inelastic cotunneling through junctions, which leads to unavoidable energy change.

### 4.1. Fundamental Bounds

The density matrix evolution of the composite quantum system representing the circuit and its surroundings depends both on physical- and logic-state changes. In order to accurately evaluate this evolution, we study the dynamics of information and identify steps of computation that lead to a change in either physical- and logic-states of any of the subsystems. In our previous work [16], we show that, for a given class of dynamically clocked circuits, a complete computational cycle of processing a given $\eta^{th}$ input may span across more than one clock cycle. For such transistor-based circuits, if each logic-state change corresponds to a distinct change in the physical system, i.e., if there is one-to-one mapping between the physical- and logic-state evolution, and if we can obtain the reduced density matrices, then we can obtain the expected energy of the bath B during a given computational step *k* as
$$\begin{array}{c} \Delta {\langle E^{B}\rangle }_{k}\geq -k_{B}T\ln\left(2\right)\times \left(\Delta I^{R_{\eta }T_{k}}+\langle \Delta S_{i}^{T_{k}}\rangle +{\langle \Delta S_{i}^{S}\rangle }_{k}+{\langle \Delta S_{i}^{D}\rangle }_{k}\right) \end{array}$$
where $I^{R_{\eta }T_{k}}$ is the amount of information erasure due to irreversible loss of correlation between the $\eta^{th}$ input referent, $R_{\eta }$, and the part of the transistor network that is active during the $k^{th}$ computational step; $\langle \Delta S_{i}^{T_{k}}\rangle$ is the expected entropy change in the associated part of the transistor network; ${\langle \Delta S_{i}^{S}\rangle }_{k}$ and ${\langle \Delta S_{i}^{D}\rangle }_{k}$ are the expected entropy changes in source, S, and drain D, associated with the given input.

In earlier circuits we analyzed, such as NASICs, under idealized conditions, the term $\langle \Delta S_{i}^{T_{k}}\rangle$ does not contribute to the fundamental lower bounds as we do not include any imperfection in the circuit operation. The transport of the electron from Source, S, to Drain, D, captures the essential functional features of computation in such circuits. In order to obtain fundamental lower bounds that are technologically relevant, we incorporate non-idealized interactions. It is important to note here that the Brownian circuits work without a clocking scheme, i.e., the analysis presented here differs from our previous work, given the asynchronous nature of the paradigm. Here, we calculate the fundamental bounds based on the difference between post-computation (final) and pre-computation (initial) states, and therefore cannot capture additional dissipation that occurs during intermediate steps. Our aim is to study the physical and computational characteristics of SET implementation of Brownian circuits on an equal footing. Therefore, our approach varies significantly from pure thermodynamic analyses of Brownian computation such as the one developed by Strasberg [40] using the stochastic Turing machine approach.

### 4.2. Free Energy Calculations

In order to determine the factors that play a key role in the performance of Brownian circuits, we use SET theory [31,32,35] and calculate free energy change and transition rate of electron tunneling, respectively, as
(1)$$\begin{array}{c} \Delta F=-q(|V_{j}|-V_{c}) \end{array}$$
(2)$$\begin{array}{c} \Gamma \left(\Delta F\right)=\Delta F/q^{2}R\left[e^{\Delta F/k_{B}T}-1\right] \end{array}$$
where *q* is electric charge, $V_{j}$ and $V_{c}$ are junction and critical voltage, respectively, *R* is junction resistance, $k_{B}$ is Boltzmann constant, *T* is temperature. Based on these equations, we calculated the free energy change in the circuit as a result of each tunneling event for all four inputs combinations. In addition to the theoretical calculations, we also used the interactive analysis on SIMON 2.0, as shown in Figure 10. A comparison between our theoretical and simulation results is laid out on Table 2.

The theoretical results for the change in free energy as a result of the computation of a single input, $\Delta F_{theory}$, is obtained by considering every single tunneling event for that input. The change in free energy for a given input, as simulated in the interactive analysis, $\Delta F_{simulation}$, is obtained by taking the difference between the final and initial free energy values. As depicted in Figure 10, the interactive analyses show the free energy value at a given tunneling step required to perform computation. Figure 10 shows that the intermediate changes in free energy are much higher than the free energy difference between the final and initial steps for a given input. This suggests that the random fluctuations add to the energy cost of computation, hinting at the difficulty of capturing dissipation involved in asynchronous paradigms. The right-most column of Table 2 represents the energy change as a result of the intermediate steps, $\Delta E_{inter}$ that take place during this asynchronous computation. It is important to note that, at T=0 K, tunneling events that lead to a decrease in free energy ΔF≤0 are allowed; therefore, we see that, for certain inputs, the simulations give us negative values. Our simulations confirm that the free energy increases during tunneling events where the tokens need to overcome a Coulomb blockade. Our analyses show that, with increasing temperature, the free energy increase reduces, since thermal energy provides some additional energy to tokens.

The comprehensive analyses we perform here combine the fundamental lower bounds introduced above with the theoretical free energy calculations, as well as the interactive analysis on SIMON 2.0. In our earlier work [23], we only focused on the lower bound of the energy required to perform computation based on the charge transport involved in token displacement and the thermodynamic cost of irreversible information erasure. However, in this study, we show that, due to the asynchronous nature of computation of Brownian circuits, the intermediate steps add to the cost of the unavoidable energy required to perform computation. Using a physical-information-theoretic approach proposed in [23,39], a generalized form of the fundamental lower bound on the energy dissipated during a complete cycle of a Brownian circuit is proposed as
(3)$$\Delta E\geq k_{B}T\ln2\Delta I+fqV_{DD}\Delta N+\Delta E_{\mathrm{interm}}$$
where *f* is a fraction of the energy required to transport ΔN many tokens with the applied bias to perform the computation. In order to obtain numerical values for this bound, we can take the drain voltage as $V_{DD}=16$ mV, the amount of tokens carrying a given input as ΔN=2 and the information irreversibly lost as ΔI=1 for the two-bit sort circuit, on average, for all possible input transition scenarios calculated similarly to that of [23]. Therefore, at T=1 K, the first term in the RHS corresponding to the thermodynamic cost of information erasure is ∼86μeV, whereas the particle supply costs in the 32 meV (for f=1) and the energy dissipated in the intermediate steps, $\Delta E_{\mathrm{interm}}$, is, on average, for all inputs, ∼53 meV; indicating that SET-based Brownian circuits cannot inherently allow us to reach $k_{B}T\ln\left(2\right)$-level dissipation, despite the reversibility inherent in its circuit primitives and randomness embedded in the workings of this computational approach.

## 5. Conclusions

Fluctuations are seen as an impediment to be avoided at any cost in future computers; however, the design of Brownian circuits exploit fluctuations. Actively searching through the state space, Brownian circuits can efficiently and reliably perform primitive logic operations in the presence of noise and fluctuations. Brownian circuit design proposes a decreased complexity for circuit primitives and topologies. As compared to earlier versions of Brownian Cellular Automata, recent designs enable the use of added circuitry to avoid the deadlocks common in conventional token-based circuits.

In the literature, SET-based Brownian cirucits are studied only for T=1 K temperature. Here, we perform physical analyses that illustrate the extreme temperature sensitivity of the Brownian circuit primitives. The Hub and CJoin elements use Brownian search in their function and can operate at higher temperatures with suitable device parameters. However, Ratchet’s unidirectional (irreversible) characteristic restricts its operation to low temperatures. Therefore, as Ratchet is necessary for obtaining accurate outputs and speeding up the circuits, our study shows that it is the key element in determining the temperature range within which a Brownian circuit can operate.

The particle supply cost dominates the fundamental lower bound in transistor-based circuits in general, far exceeding the cost of the irreversible logic operations. At cryogenic temperatures, $k_{B}T\ln\left(2\right)$ becomes negligible. Implementations based on SET technology promise lower fundamental limits compared to other transistor-based circuits; however, this technology-base is restricted to low-temperature operations for Brownian circuits. Furthermore, the area required for Brownian circuits is small, despite the dual-rail encoding; however, the time required for computation is long. Therefore, power dissipation may be relatively high.

The manufacturing techniques proposed for SET technology allow aggressive scaling, which can mean higher performance, density, and power efficiency that can go far beyond the performance of CMOS technology. Employing an approach that incorporates physical interactions embedded in asynchoronous circuit operation is the key in revealing tighter fundamental bounds, as illustrated in contributions by inelastic tunneling events. In short, despite the reversibility and randomness embedded in Brownian circuits, our results indicate that we cannot reach $k_{B}T\ln\left(2\right)$-level dissipation in SET-based implementations of Brownian circuits.

## Figures and Tables

**Figure 1 entropy-23-00406-f001:**
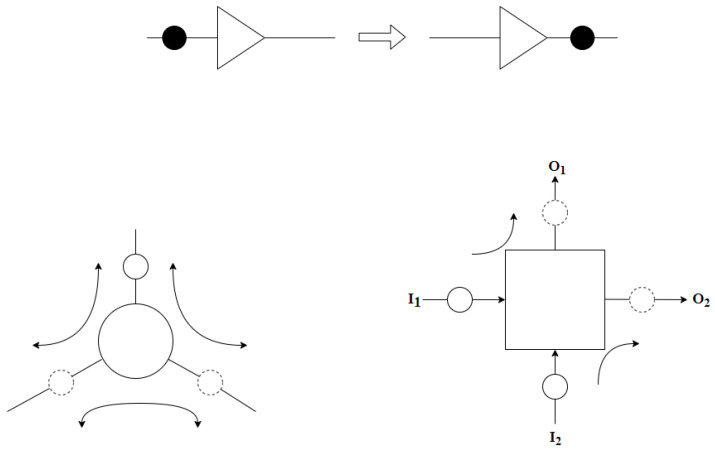
Brownian circuit primitive Ratchet (**top**), Hub (**left**) and CJoin ( **right**), where the inputs and outputs are designated based on the token follow direction shown in the figure.

**Figure 2 entropy-23-00406-f002:**
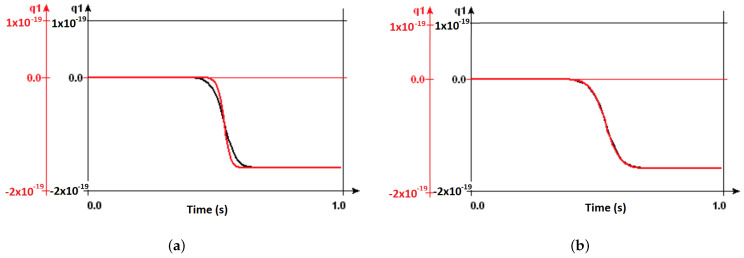
Tunneling behaviour of a Brownian Ratchet at T=4 K (**a**) and 9 K (**b**) for $J_{1}=J_{2}=10^{-17}$ F (black line), to $J_{1}=J_{2}=10^{-18}$ F (red line).

**Figure 3 entropy-23-00406-f003:**
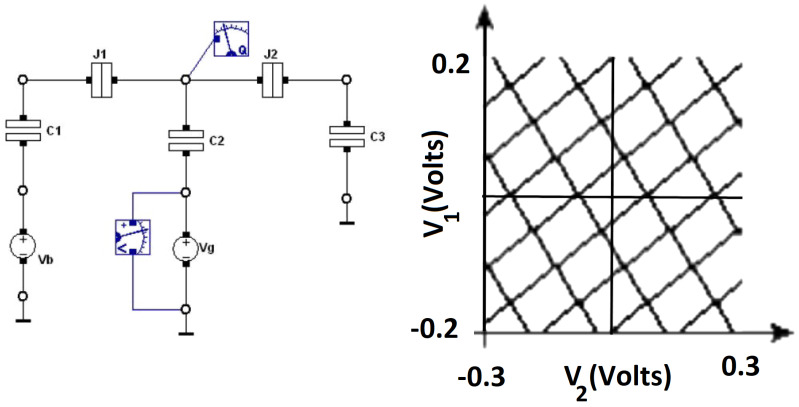
A Brownian Ratchet (**left**), as depicted in SIMON 2.0, and its stability plot with respect to bias voltage $V_{b}=V_{1}$ and gate voltage $V_{g}=V_{2}$ (**right**).

**Figure 4 entropy-23-00406-f004:**
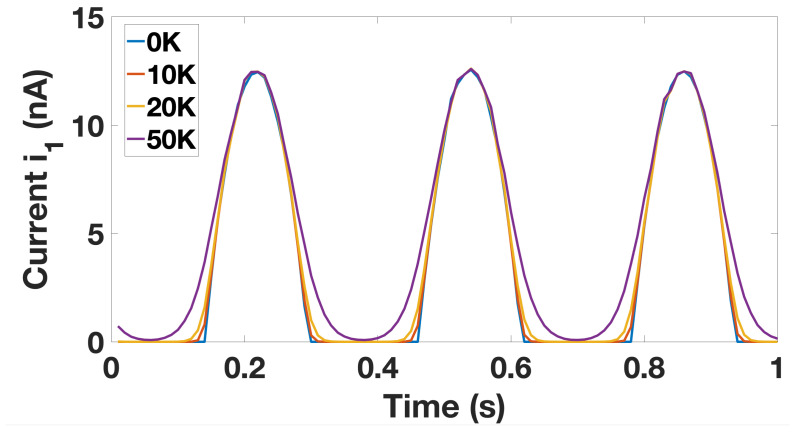
The change in current as a function of temperature.

**Figure 5 entropy-23-00406-f005:**
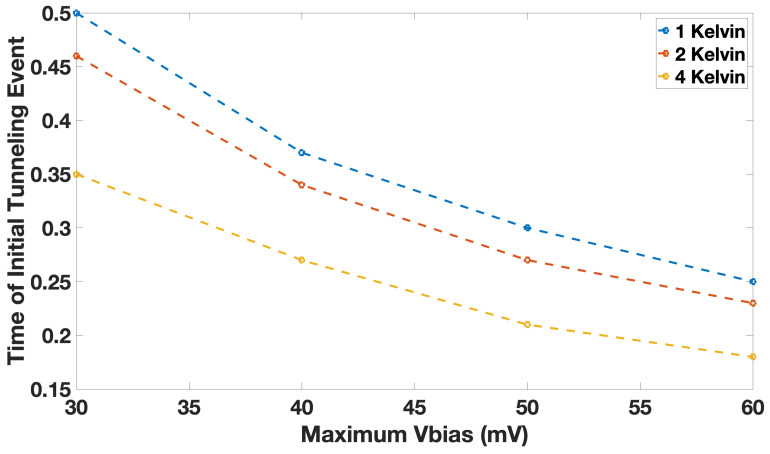
The time at which the first tunneling event occurs changes as a function of bias voltage $V_{bias}$ and temperature T=1,2 and 4 K, for $V_{gate}=V_{bias}$.

**Figure 6 entropy-23-00406-f006:**
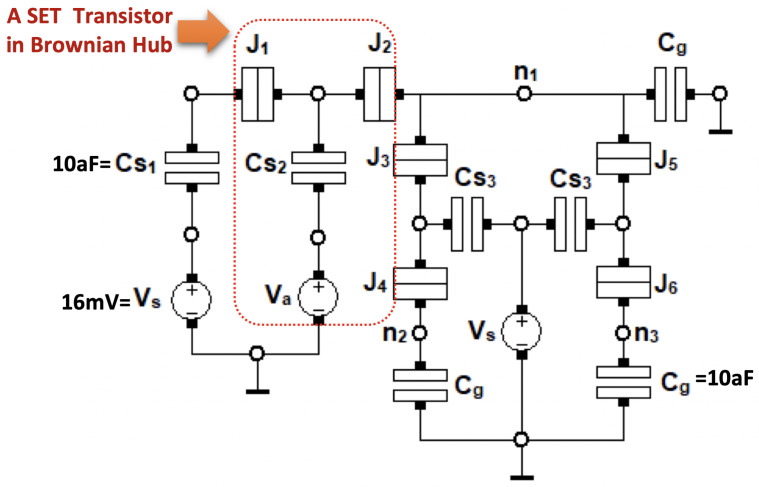
SET-based Brownian Hub circuit element.

**Figure 7 entropy-23-00406-f007:**
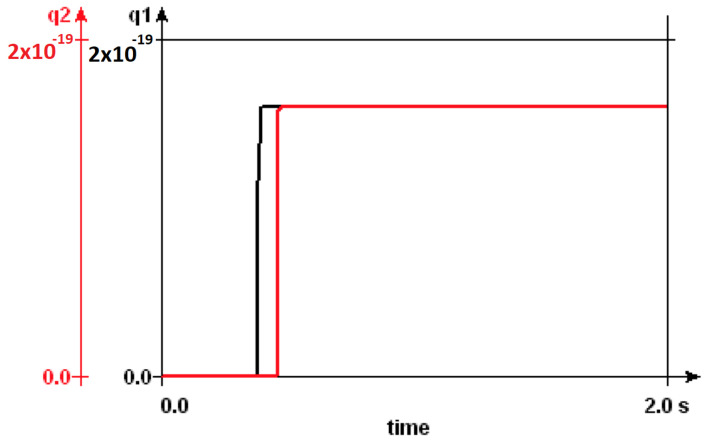
Net charge (in Coulombs) at the outputs of a CJoin at *T* = 1 K, 5 K and 50 K illustrating the increase in fluctuations with temperature.

**Figure 8 entropy-23-00406-f008:**
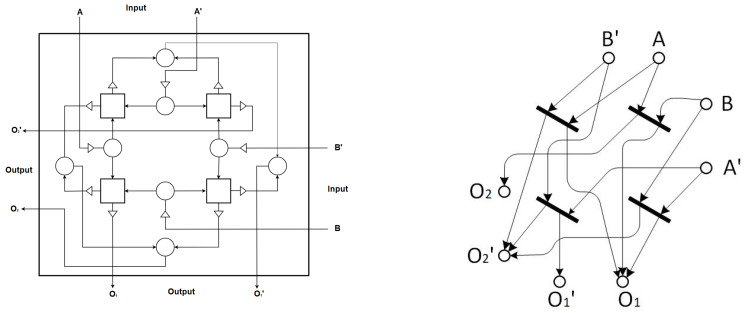
Brownian two-bit sort (**left**) constructed from four CJoins (squares), eight Hubs (circles) and twelve Ratchets (triangles) and its Petri net (**right**).

**Figure 9 entropy-23-00406-f009:**
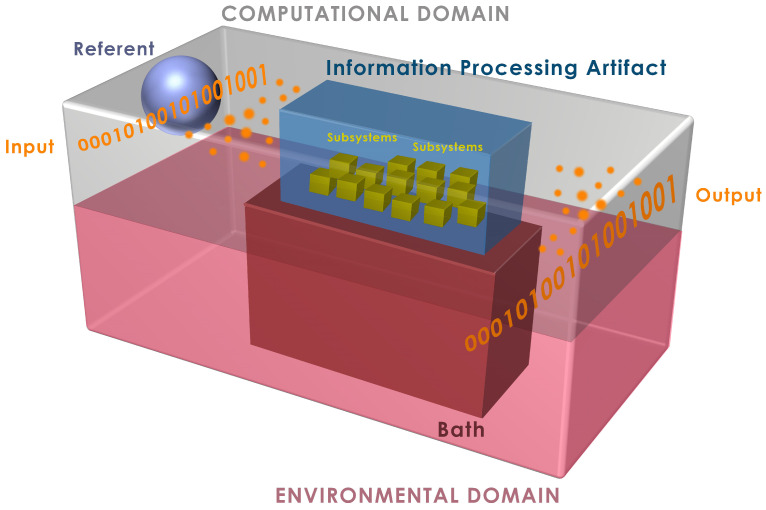
Physical abstraction of the information processing artifact and its surrounding subsystems situated in a globally closed and isolated universe based on [16].

**Figure 10 entropy-23-00406-f010:**
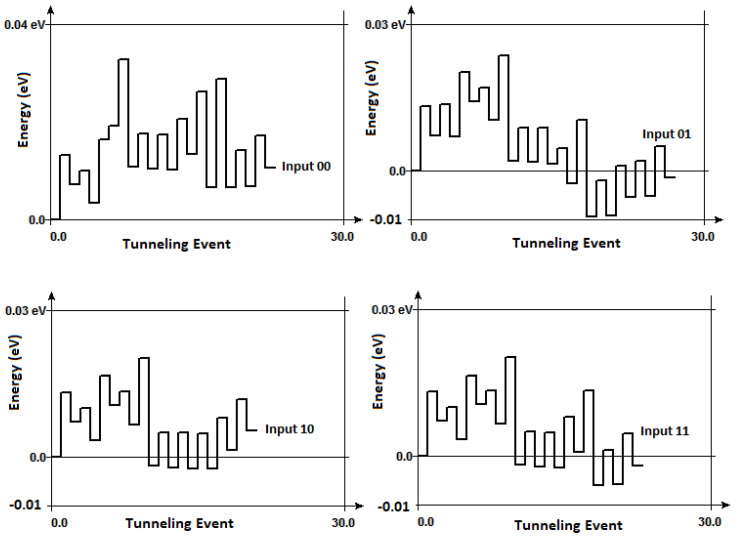
Interactive analysis results from SIMON 2.0 of the Brownian Two-Bit Sort Circuit for all four inputs as a function of each tunneling event required to process for input-00 (**top left**), -01 (**top right**), -10 (**bottom left**), -11 (**bottom right**).

**Table 1 entropy-23-00406-t001:** The truth table of the Brownian two-bit sort presented in Figure 8 with inputs A, B and output O, along with their complements.

INPUTS				OUTPUTS			
A	$A^{\prime }$	B	$B^{\prime }$	$O_{1}$	$O_{1}^{\prime }$	$O_{2}$	$O_{2}^{\prime }$
1	0	1	0	1	0	1	0
1	0	0	1	1	0	0	1
0	1	1	0	1	0	0	1
0	1	0	1	0	1	0	1

**Table 2 entropy-23-00406-t002:** Input dependence of tunneling events and associated free energy values for the SET-based Brownian two-bit sort circuit.

IN	# of Tunneling Events	$\Delta F_{\mathrm{theory}}$ (meV)	$\Delta F_{\mathrm{simulation}}$ (meV)	$\Delta E_{\mathrm{inter}}$ (meV)
00	22	51.49	10.76	40.73
01	26	69.73	−1.25	70.98
10	22	52.06	5.53	46.53
11	22	51.78	−1.97	53.75
Average	23	56.27	3.27	53.00

## Data Availability

The data presented in this study are available on request from the corresponding author. The data are not publicly available due to multiple licensed software used that were sponsored by different institutions under various grants.

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
