# Peer review of "Physical Limitations on Fundamental Efficiency of SET-Based Brownian Circuits"

_entropy, 2021, doi:10.3390/e23040406_

Round 1

Reviewer 1 Report

The paper is well-written and provide an interesting discussion of the limitations of SET-Based Brownian Circuits. However, there are some minor points that should be checked and corrected:

1) The author should use the same format for citing paper of the bibliography (for some papers they use Ref. and not for others). Also, they should use [1] for quoting Peper et al. 

2) In pag. 1 row 32 "we can" is repeated twice.

3) Figure 1. The position of the subplots is not correct (bottom, bottom left, ...)

Author Response

We would like the thank you and our reviewers for considering our work on ``Physical Limitations on Fundamental Efficiency of SET-Based Brownian Circuits" presented in entropy-1113339 manuscript.  We find the detailed comments provided by our reviewers to be extremely helpful and insightful. We would like to thank our reviewers for taking the time to  reflect on points that will help strengthen the impact of our study and make it more accessible for the scientific community. 
We have carefully considered each and every suggestion presented by our reviewers and updated the manuscript to the best of our ability. In the attached PDF, we outline the changes made on our manuscript in response to each comment by our reviewers.

Reviewer 2 Report

The authors present a study of the physical conditions that characterise the efficiency of so-called Brownian circuits and discuss a possible implementation of these circuits with Single Electron Transistors.

Their analysis represents a specific application of a previously developed (by the same authors) general approach for logic circuits.

The ms might be interesting for the vast community of researchers interested in fundamental limits of computing devices; even if there are parts that appear to be redundant and, frankly, unnecessary. Like, for example, the discussion of the functioning of the simulation tools SIMON 2.0 for ratchets.

Specifically, there are few relevant points that should be addressed, before a favourable recommendation can be formulated.

1) The introduction of the paper is very poorly presented. The authors introduce Brownian circuits as a novel technology to efficiently and reliably perform primitive logic operations in the presence 
of noise and fluctuations and list three recent references.

The authors should mention that the role of fluctuations in operating logic gates is certainly not new nor is limited to the cited references.

For their reference, I list here some papers that might help the authors to better put their introduction in a right perspective:
- Pinar Korkrnaz,  Bilge E. S. Akgul,  Krishna V. Palem, Ultra-Low Energy Computing with Noise: Energy-Performance-Probability Trade-offs, Proceedings of the 2006 Emerging VLSl Technologies and Architectures (ISVLSISO6), 2006

- K. Murali, Sudeshna Sinha, William L. Ditto, and Adi R. Bulsara, Reliable Logic Circuit Elements that Exploit Nonlinearity in the Presence of a Noise Floor, Phys. Rev. Lett. 102, 104101 – Published 10 March 2009

- L. Worschech, F. Hartmann, T. Y. Kim, et al,  Universal and reconfigurable logic gates in a compact three-terminal resonant tunneling diode, Appl. Phys. Lett. 96, 042112 (2010)

- Diego N. Guerra, Adi R. Bulsara, William L. Ditto et al, A Noise-Assisted Reprogrammable Nanomechanical Logic Gate, Nano Lett. 2010, 10, 4, 1168–1171, March 10, 2010

- Anna Dari, Behnam Kia, Adi R. Bulsara, and William L. Ditto, Logical stochastic resonance with correlated internal and external noises in a synthetic biological logic block, Chaos 21, 047521 (2011)

And, on the quantum side:

- L.-M. Duan and R. Raussendorf, Efficient Quantum Computation with Probabilistic Quantum Gates, Phys. Rev. Lett. 95, 080503 – Published 17 August 2005

- Dorit Aharonov, Alexei Kitaev, and John Preskill, Fault-Tolerant Quantum Computation with Long-Range Correlated Noise, Phys. Rev. Lett. 96, 050504 – Published 7 February 2006

Just to mention a few.

2) Line 31. The authors write: “There is an ongoing effort to assess fundamental limitations of novel circuit paradigms, to help assess upcoming trends in future of computing [5].”

In addition to ref. 5 it is probably useful for the reader to check also:

- K. Natori and N. Sano, Scaling limit of digital circuits due to thermal noise. J. of Applied Physics

- L. Gammaitoni, Noise limited computational speed, Appl. Phys. Lett. 91, 224104 (2007)

3) Line 32. The authors write: “Recent experiments show that we can we can obtain energy dissipation in the kBT-level.” Apart from the repetition, that should be eliminated, a reference is needed here. 

4) Line 51 and 54. The world “effect” should be “affect”.

5) Line 66. Authors write: “As compared to Hub and CJoin, ratchet performs an irreversible operation.” What do they mean? Do they mean “asymmetric”? What kind of irreversibility are we talking here?

6) In general, the ms could benefit from a revision of the English. As an example, in Line 67. There is something wrong in this sentence: “For a technology to be suitable for implementations of Brownian circuits, it is necessary that it supports representation of signals that is token-based and that are subject to fluctuations universal logic operations can be obtained based on the transition rules associated with each element and can be implemented using SET technology.”

Another example is Line 205: “When we apply an input to Brownian two-bit sort circuit, all components do not work.” 

7) When the ms enters the original part of the contribution of the authors, unfortunately, little effort is spent to put the reader in condition to easily follow the proposed reasoning. As an example, in Line 216 the authors write: “…the irreversible information loss with loss of correlation between the computing system and the referent holding memory of the initial input.” This sentence is quite obscure to me. Why the loss of coherence is also a loss of information. What do they mean?

Another sentence that is not clear enough is at Line 277 (and following): “ However in this study, we show that due to the asynchronous nature of computation of Brownian circuits, the intermediate steps add to the cost of the unavoidable energy required to perform computation. ” Why is the “asynchronous nature of computation” responsible for that? Probably this point deserves a better explanation.

Author Response

(The authors gave the same response as above.)

Round 2

Reviewer 2 Report

The authors have significantly improved the presentation of their findings.
However, there is a point that still deserves some attention.

In the new versione of the ms they write: 
“As compared to Hub and CJoin primitives, ratchet performs asymmetric operation, allowing tokens to flow only in forward direction; tokens are not allowed to tunnel backwards, which is an illustration of thermodynamic irreversibility leading to deadlocks, a phenomenon commonly seen in conventional token based circuits. Based on the functional description of these primitives, ...”

It seems to me that the fact that " tokens are not allowed to tunnel backwards" it is an illustration (at most) of logic irreversibility. The association with thermodynamic irreversibility is not granted.
As the authors know logic irreversibility does not necessarily implies thermodynamic irreversibility.
This point should be addressed.

Author Response

Dear Reviewer, 

We would like to thank you for your invaluable feedback. The point you make is  a crucial one and after careful consideration we decided that it is best to leave out any discussion that may lead our readers to confusion. Therefore we updated the sentence of concern as follows:

"As compared to Hub and CJoin primitives,  Ratchet performs  asymmetric operation, allowing tokens to flow only in forward direction;  tokens are not allowed to tunnel backwards, which leads to deadlocks, a phenomenon commonly seen in conventional token based circuits. Based on the functional description...."

 We believe that the discussion we make later on on the asymmetric and unidirectional operation of Ratchet and its relation to logic reversibility will be sufficient. We left out the thermodynamic irreversibility discussion since that is not the main focus of our manuscript. We hope that this revision addresses the point you raised. 

In addition to this minor revision, we also fixed minor typos suchs as "we shows" and "rachet," and we also wrote all "Ratchet"s in capital as it appears in the literature for consistency. We also eliminated the alternating usage of "Single Electron Tunneling (SET) Transistor" and "Single Electron Transistor," and simply kept the latter. Similarly, "two bit sort" and "two bit sorter" usage is fixed by replacing all instances of the phrase with the former. Lastly, we fixed minor formatting errors in the bibliography and updated the acknowledgements section. 

We are deeply grateful for your careful consideration of our work and  look forward to hearing your feedback if there is any other points you think we should address. 

Kind regards, 

Dr. Ilke Ercan 

Round 3

Reviewer 2 Report

The authors have addressed  the point I raised.
The ms can now be published.